# Early antiretroviral therapy for HIV-infected patients admitted to an intensive care unit (EARTH-ICU): A randomized clinical trial

**Márcio M. Boniatti**[1]*, **José Augusto S. Pellegrini**[1], **Leonardo S. Marques**[2], **Josiane F. John**[1], **Luiz G. Marin**[2], **Lina R. D. M. Maito**[3], **Thiago C. Lisboa**[4], **Lucas P. Damiani**[5], **Diego R. Falci**[6]

1 Critical Care Department, Hospital de Clínicas de Porto Alegre, Universidade La Salle, Porto Alegre, Brazil,
2 Critical Care Department, Hospital Nossa Senhora da Conceição, Porto Alegre, Porto Alegre, Brazil,
3 Critical Care Department, Hospital São Vicente de Paulo, Passo Fundo, Brazil, 4 Critical Care Department, Hospital de Clínicas de Porto Alegre, Instituto de Pesquisa HCor, Universidade La Salle, Porto Alegre, Brazil, 5 Instituto de Pesquisa HCor, Sao Paulo, Brazil, 6 Infectious Disease Department, Hospital de Clínicas de Porto Alegre, Porto Alegre, Brazil

* mboniatti@hcpa.edu.br

## Abstract

### Background

Highly active antiretroviral therapy (HAART) has reduced HIV-related morbidity and mortality at all stages of infection and reduced transmission of HIV. Currently, the immediate start of HAART is recommended for all HIV patients, regardless of the CD4 count. There are several concerns, however, about starting treatment in critically ill patients. Unpredictable absorption of medication by the gastrointestinal tract, drug toxicity, drug interactions, limited reserve to tolerate the dysfunction of other organs resulting from hypersensitivity to drugs or immune reconstitution syndrome, and the possibility that subtherapeutic levels of drug may lead to viral resistance are the main concerns. The objective of our study was to compare the early onset (up to 5 days) with late onset (after discharge from the ICU) of HAART in HIV-infected patients admitted to the ICU.

### Methods

This was a randomized, open-label clinical trial enrolling HIV-infected patients admitted to the ICU of a public hospital in southern Brazil. Patients randomized to the intervention group had to start treatment with HAART within 5 days of ICU admission. For patients in the control group, treatment should begin after discharge from the ICU. The patients were followed up to determine mortality in the ICU, in the hospital and at 6 months. The primary outcome was hospital mortality. The secondary outcome was mortality at 6 months.

### Results

The calculated sample size was 344 patients. Unfortunately, we decided to discontinue the study due to a progressively slower recruitment rate. A total of 115 patients were randomized. The majority of admissions were for AIDS-defining illnesses and low CD4. The main

**Data Availability Statement:** We made the manuscript data available through file upload (as Supporting Information file).

**Funding:** The author(s) received no specific funding for this work.

**Competing interests:** The authors have declared that no competing interests exist.

**Abbreviations:** AIDS, Acquired immunodeficiency syndrome; ARDS, Acute respiratory distress syndrome; CMV, Cytomegalovirus; CP, Conditional power; HAART, Highly active antiretroviral therapy; HIV, Human immunodeficiency virus; ICU, Intensive Care Unite.

cause of admission was respiratory failure. Regarding the early and late study groups, there was no difference in hospital (66.7% and 63.8%, p = 0.75) or 6-month (68.4% and 79.2%, p = 0.20) mortality. After multivariate analysis, the only independent predictors of in-hospital mortality were shock and dialysis during the ICU stay. For the mortality outcome at 6 months, the independent variables were shock and dialysis during the ICU stay and tuberculosis at ICU admission.

## Conclusions

Although the early termination of the study precludes definitive conclusions being made, early HAART administration for HIV-infected patients admitted to the ICU compared to late administration did not show benefit in hospital mortality or 6-month mortality.

ClinicalTrials.gov, NCT01455688. Registered 20 October 2011, https://clinicaltrials.gov/show/NCT01455688

## Introduction

Highly active antiretroviral therapy (HAART) has changed the natural history of patients infected with human immunodeficiency virus (HIV). Before HAART's advent, the disease was potentially fatal. After the introduction of HAART, HIV began to acquire the characteristics of chronic disease, with a substantial improvement in survival [1–3]. The rates of intensive care unit (ICU) admission have also changed, especially for the spectrum of diseases not associated with acquired immunodeficiency syndrome (AIDS) as an initial diagnosis [4–6]. Although such diseases are becoming more common, opportunistic infections are still the main cause of hospitalization in intensive care, especially in poor countries or among patients with low adherence to treatment or those who have difficulty accessing the health system [7, 8].

HAART has reduced HIV-related morbidity and mortality at all stages of infection and reduced transmission of HIV [9–16]. Two large clinical trials evaluating the optimal timing to start HAART demonstrated an approximately 50% reduction in morbidity and mortality among subjects with HIV and CD4+ > 500 cells / mm3, randomized to receive HAART immediately compared to late onset [10, 11]. Currently, the immediate start of HAART is recommended for all HIV patients, regardless of the CD4 count [17]. There are several concerns, however, about starting treatment in critically ill patients. Unpredictable absorption of medication by the gastrointestinal tract, drug toxicity, drug interactions, limited reserve to tolerate the dysfunction of other organs resulting from hypersensitivity to drugs or immune reconstitution syndrome (IRIS), and the possibility that subtherapeutic levels of drug may lead to viral resistance are the main concerns [18]. Some retrospective studies have found a decrease in mortality among patients with HAART in the ICU [19, 20], while others have failed to show this difference [21, 22]. A recent meta-analysis suggested increased survival for patients treated with HAART during ICU admission [23]. To date, however, no randomized clinical trial has been conducted to test this hypothesis. Thus, the objective of our study was to compare, in relation to hospital mortality, the early onset (up to 5 days) with late onset (after discharge from the ICU) of HAART in HIV-infected patients admitted to the ICU.

## Material and methods

This was a randomized, open-label clinical trial enrolling HIV-infected patients admitted to the ICU of a public hospital in southern Brazil between January 2012 and December 2015.

Patients were consecutively evaluated for inclusion if they were older than 18 years and had 1) CD4 cell count lower than 350 cells/ml in the 3 months prior to the study, 2) CD4 cell count between 350 cells/ml and 500 cells/ml and age > 55 years, coinfection with hepatitis B virus or hepatitis C virus, neoplasia, viral load > 100,000 copies/ml or elevated cardiovascular risk, or 3) an AIDS-defining disease, which were the criteria required for HIV treatment indication in Brazil, during the period of study. Patients with regular use of HAART, pregnancy, impossibility to use the enteral route, tuberculous meningitis, cryptococcal meningitis, ICU length of stay > 5 days prior to randomization or who refused to sign the informed consent were excluded from the study.

This study protocol was consistent with the ethical principles of the Declaration of Helsinki and was previously approved by an institutional committee of research ethics. Written informed consent was obtained from each included patient or from the next of kin. The study protocol was recorded in the ClinicalTrials.gov database (NCT01455688) prior to initiation.

The study was conducted in a clinical-surgical ICU of a tertiary public hospital in southern Brazil. The hospital has 843 beds, with approximately 26,000 hospitalizations per year. The ICU has 59 beds, of which 14 are postoperative beds used after major surgery, and the remaining beds are clinical beds.

Randomization was performed by random generation of sequences by software (www. randomizer.org) in a ratio of 1:1. Randomization in blocks of 10 was performed and stratified according to the SAPS III. Concealment of allocation was ensured by sealed, opaque, sequentially numbered envelopes.

Patients randomized to the intervention group had to start treatment with HAART within 5 days of ICU admission. For patients in the control group, treatment should begin after discharge from the ICU. The HAART scheme was determined by a hospital infectologist accompanying HIV-infected patients admitted to the ICU. All other interventions were defined by the assistant team.

Data included the age, sex, reason for admission to the ICU, location prior to admission to the ICU (emergency, ward or other hospital) and SAPS III. Admission was considered AIDS-related if the main admission diagnosis was an AIDS-defining illness [24]. The following therapeutic modalities in the ICU were recorded: invasive mechanical ventilation, hemodialysis or hemofiltration and the use of vasoactive drugs. The following data on HIV infection were recorded for each case at ICU admission: CD4 cell count and viral load, if available within 3 months of admission, and opportunistic infections. The following HAART data were recorded: scheme chosen, time to onset, adverse effects and need for suspension or modification of the regimen. HAART has been defined as a combination of 3 or more antiretrovirals belonging to at least two classes of the following: protease inhibitors, nucleoside reverse transcriptase inhibitors and non-nucleoside reverse transcriptase inhibitors, according to Brazilian guidelines at time of diagnosis.

The diagnosis of pneumocystosis was confirmed if *P. jirovecii* was identified in the bronchoalveolar lavage with silver staining. The diagnosis of tuberculosis was based on identification of the agent in tissues or sputum with acid-fast staining. The presumptive diagnosis of pneumocystosis or tuberculosis was made if the patient had a compatible medical history and physical examination, and the assistant team started treatment. Cryptococcal meningitis was diagnosed with India's positive ink test for fungi, a positive cerebrospinal fluid culture or a positive cryptococcal antigen detection test. The presumptive diagnosis of cerebral toxoplasmosis was accepted if the patient presented with a lesion with a mass effect and/or contrast impregnation in the cranial computed tomography associated with neurological alteration. Diagnosis of cytomegalovirus (CMV) infection was performed through CMV serology (IgG) plus histopathological findings in pulmonary or gastrointestinal biopsies. Shock was defined as

the need for vasopressors to maintain a mean arterial pressure > 65 mmHg. Acute Respiratory Distress Syndrome (ARDS) was defined as respiratory dysfunction characterized by acute onset, bilateral opacities on chest radiography and a PaO2 / FiO2 ratio ≤ 300 with final expiratory pressure or continuous positive airway pressure ≥ 5 cmH$_2$O [25].

The patients were followed up to determine mortality in the ICU, in the hospital and at 6 months. The primary outcome was hospital mortality. The secondary outcome was mortality at 6 months.

## Statistical analysis

The sample was calculated to verify a reduction in hospital mortality of 15%, considering 55% of mortality in the control group, with a significance level 5% and study power 80%. The calculated sample size was 344 patients (172 patients in each group). As we did not reach the calculated sample size, we performed the conditional power (CP) computation *a posteriori*. CP is the probability that the final study result will be statistically significant, given the data observed thus far and a specific assumption about the pattern of the data to be observed in the remainder of the study, such as assuming the original design effect, or the effect estimated from the current data, or under the null hypothesis. In many clinical trials, a CP computation at a pre-specified point in the study, such as mid-way, is used as the basis for early termination for futility when there is little evidence of a beneficial effect [26]. We decided to discontinue the study due to a progressively slower recruitment rate. Nevertheless, the calculated conditional power assuming the original estimated effect was 49.5%, and the observed effect from the current data was only 3.2%, suggesting futility.

Continuous variables were described as means ± standard deviation or medians and inter-quartile range, according to normality criteria. Categorical variables were shown as frequencies and proportions. Differences between groups at baseline were analyzed with a t-test for two independent samples or the Wilcoxon-Mann-Whitney U test according to normality criteria. Fisher's exact test was applied to the categorical variables.

The primary analysis was made by unadjusted intention-to-treat comparisons between the two study groups in relation to the primary and secondary outcomes. To examine the association between early or late onset of HAART and other variables of interest with hospital mortality and mortality at 6 months, a univariate logistic analysis was performed with hospital mortality and mortality at 6 months as dependent variables. A multivariate model was constructed to identify variables independently associated with hospital mortality and mortality at 6 months. For multivariate analysis, early or late initiation of HAART was maintained as a variable of interest. Other variables defined a priori were variables previously reported in the literature with an association to these outcomes or those plausibly associated. In addition, we included any variable that resulted in a value of p <0.20 in the univariate analysis. The regression was then constructed through stepwise forward, excluding the variables without significant association in each step, until the final model. All analyses were performed using IBM SPSS Statistics, version 20.0 (IBM corp., Armonk, NY, USA) and R (version 3.6.0). Statistical significance was set at 0.05.

## Results

A total of 268 HIV-infected patients were admitted to the ICU during the study period. Of these, 153 were excluded, mainly because of previous regular use of HAART. Thus, 115 patients were randomized (Fig 1). In December 2015, we decided to discontinue the study due to a progressively slower recruitment rate in the previous 12 months. Follow-up of patients was extended until June 2016 to verify 6-month mortality of included patients. The

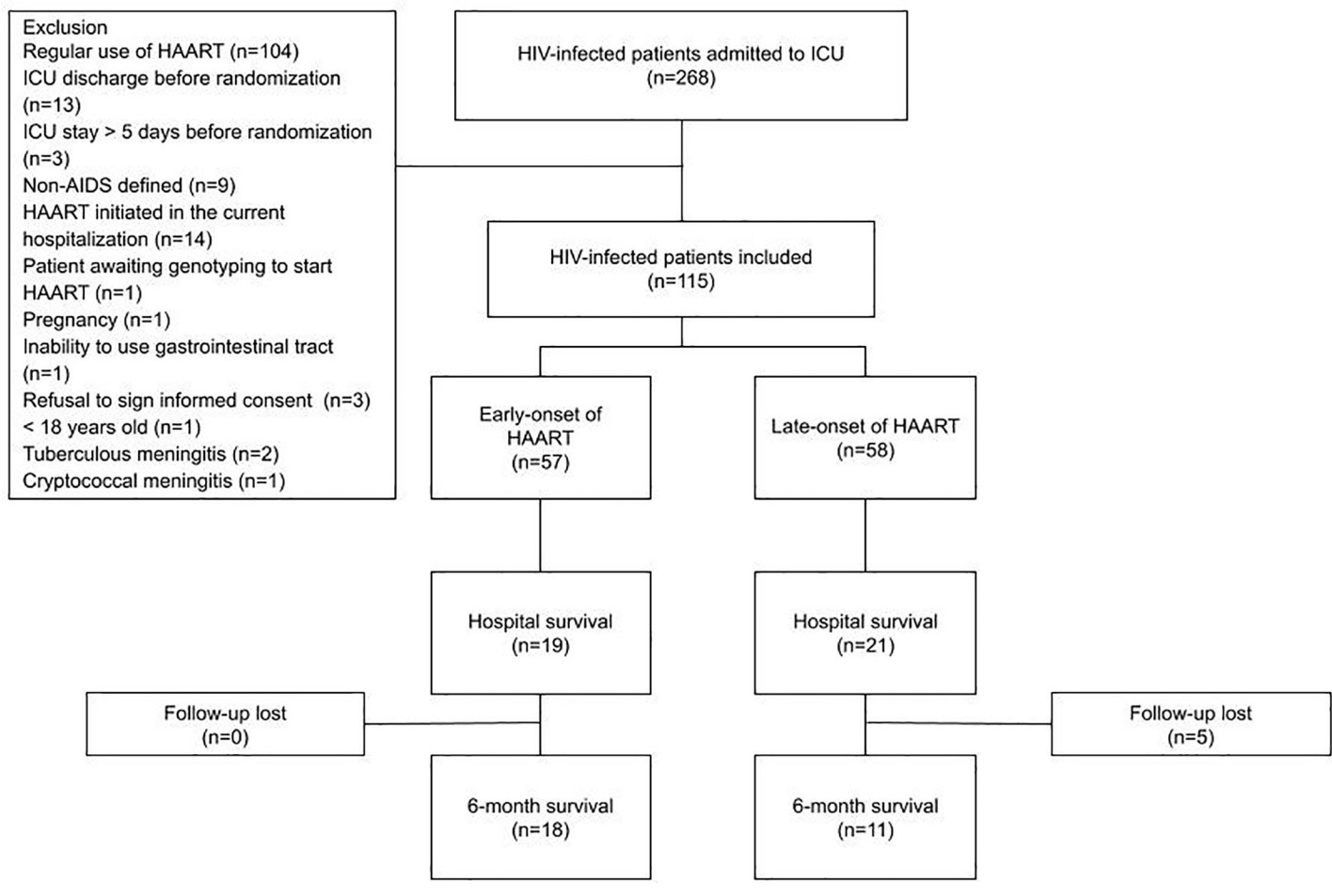

**Fig 1. Flowchart for inclusion of patients in the study.** HAART, highly active antiretroviral therapy; ICU, Intensive Care Unit.

sociodemographic and clinical characteristics of patients are described in Table 1. The majority of admissions were for AIDS-defining illnesses and low CD4+ cell counts (median <50 cells / ml). The main cause of admission was respiratory failure. Nearly half of the patients had tuberculosis or pneumocystosis. All patients required mechanical ventilation during the ICU stay.

By univariate analysis, the factors associated with hospital mortality were admission from the ward, SAPS III, shock and dialysis. In multivariate analysis, the only variables associated with in-hospital mortality, after controlling for other factors, were shock and dialysis during the ICU stay (Table 2). At 6 month, variables associated to mortality in multivariate analysis were shock and dialysis during the ICU stay and diagnosis of tuberculosis at ICU admission.

The patients' outcomes according to the group are described in Table 3. The primary outcome was not different between the groups. Also, there was no difference in length of hospital stay or in ICU and 6-month mortality.

Regarding the two study groups, the strategies for initiating HAART were different. The median number of days for initiation of HAART from ICU admission were 3.0 (2.0–4.0) days and 21.0 (16.5–28.0) days in the early-onset and late-onset groups, respectively (p <0.001). In the early-onset group, 4 (7.0%) patients did not initiate HAART because of hemodynamic instability and subsequent death prior to start of treatment. Seven (12.3%) had delayed onset

**Table 1. Demographic and clinical variables.**

| Variables | Early onset (n = 57) | Late-onset (n = 58) | p |
|---|---|---|---|
| Sex, male, n (%) | 33 (57.9) | 35 (60.3) | 0.79 |
| Age, years, mean ± SD | 43.5 ± 10.3 | 45.6 ± 12.3 | 0.33 |
| HIV diagnosis, month, median (IQR) | 2.0 (0–48.0) | 7.5 (0–72.0) | 0.62 |
| Source, n (%) | | | 0.70 |
| Emergency | 28 (49.1) | 33 (56.9) | |
| Ward | 28 (49.1) | 24 (41.4) | |
| Another hospital | 1 (1.8) | 1 (1.7) | |
| Length of stay prior to ICU admission, days, median (IQR) | 4.0 (2.0–10.5) | 4.0 (2.0–9.0) | 0.69 |
| Cause of admission, n (%) | | | 0.86 |
| Respiratory failure | 33 (57.9) | 40 (69.0) | |
| Sepsis | 12 (21.1) | 8 (13.8) | |
| Neurologic | 8 (14.0) | 7 (12.1) | |
| Gastrointestinal | 1 (1.8) | 1 (1.7) | |
| Cardiovascular | 2 (3.5) | 1 (1.7) | |
| Postoperative | 1 (1.8) | 1 (1.7) | |
| AIDS-related illness, n (%) | 38 (66.7) | 42 (72.4) | 0.50 |
| SAPS III, mean ± SD | 72.4 ± 13.0 | 73.0 ± 12.5 | 0.81 |
| CD4 count, median (IQR) | 37,0 (15.0–106.25) | 48.5 (16.0–128.25) | 0.54 |
| Viral load, median (IQR) | 124,935.0 (25,897.0–475,336.5) | 186,418.0 (33,745.5–500,000.0) | 0.54 |
| Tuberculosis, n (%) | 23 (40.4) | 28 (48.3) | 0.39 |
| Cryptococcosis, n (%) | 3 (5.3) | 7 (12.1) | 0.19 |
| Cytomegalovirus infection, n (%) | 9 (15.8) | 7 (12.1) | 0.56 |
| Toxoplasmosis, n (%) | 7 (12.3) | 3 (5.2) | 0.18 |
| Pneumocystosis, n (%) | 27 (47.4) | 33 (56.9) | 0.31 |
| Mechanical ventilation, n (%) | | | |
| On admission | 49 (86.0) | 52 (89.7) | 0.55 |
| During ICU stay | 57 (100) | 58 (100) | - |
| Dialysis, n (%) | 26 (45.6) | 20 (34.5) | 0.22 |
| Shock, n (%) | 43 (75.4) | 46 (79.3) | 0.62 |
| ARDS, n (%) | 21 (36.8) | 23 (39.7) | 0.76 |

(beginning after 5 days). In the late onset group, 37 (63.8%) patients did not initiate HAART during the hospital stay. The most common antiretroviral regimens used were tenofovir + lamivudine + efavirenz (n = 26; 35.1%), abacavir + lamivudine + efavirenz (n = 21, 28.4%) and zidovudine + lamivudine + efavirenz (n = 17, 23.0%). Regardless of the group, 11 (9.6%)

**Table 2. Outcomes.**

| Outcome variables | Early onset (n = 57) | Late-onset (n = 58) | p |
|---|---|---|---|
| Length of ICU stay, days, median (IQR) | 16.0 (9.0–27.0) | 14.0 (10.0–20.25) | 0.61[a] |
| Length of hospital stay, days, median (IQR) | 32.0 (18.25–54.0) | 31.0 (21.0–53.5) | 0.97[a] |
| ICU mortality, n (%) | 29 (50.9) | 26 (44.8) | 0.52[b] |
| Hospital mortality, n (%) | 38 (66.7) | 37 (63.8) | 0.75[b] |
| 6-month mortality, n (%) | 39/57 (68.4) | 42/53 (79.2) | 0.20[b] |

[a] Mann-Whitney U Test.

[b] Chi-square test.

Table 3. Univariate and multivariate analyses of variables associated with hospital mortality and 6-month mortality[a].

| Variables | Hospital Mortality | | Six-month Mortality | |
|---|---|---|---|---|
| | Crude OR | Adjusted OR | Crude OR | Adjusted OR |
| Age | 1.03 (0.99–1.07) | | 1.03 (0.99–1.07) | |
| Early onset HAART | 0.88 (0.41–1.90) | | 1.76 (0.74–4.20) | |
| Source, ward | 2.25 (1.01–5.02) | | 2.39 (0.97–5.88) | |
| AIDS-related illness | 0.97 (0.42–2.24) | | 0.85 (0.33–2.19) | |
| Tuberculosis | 2.13 (0.96–4.76) | | 4.33 (1.60–11.78) | 4.00 (1.34–11.95) |
| Pneumocystosis | 0.72 (0.33–1.56) | | 0.96 (0.41–2.23) | |
| Cryptococcosis | 5.32 (0.65–43.58) | | - | |
| SAPS III | 1.04 (1.01–1.08) | | 1.05 (1.01–1.09) | |
| CD4 count | 1.00 (0.99–1.00) | | 0.99 (0.99–1.00) | |
| Sepsis | 3.61 (0.99–13.20) | | 3.58 (0.77–16.6) | |
| Shock | 5.42 (2.12–13.84) | 3.60 (1.25–10.31) | 5.36 (2.07–13.90) | 3.16 (1.06–9.37) |
| Dialysis | 5.11 (2.01–12.98) | 3.14 (1.13–8.73 | 9.33(2.61–33.3) | 5.31 (1.35–20.87) |
| Mechanical ventilation on ICU admission | 0.72 (0.21–2.47) | | 1.14 (0.33–3.95) | |
| ARDS | 0.89 (0.41–1.96) | | 0.88 (0.37–2.08) | |

[a] Logistic regression analysis.

patients had antiretroviral adverse effects: hepatotoxicity (n = 3, 2.6%), nephrotoxicity (n = 2, 1.7%) and hematological dysfunction (n = 6; 5.2%). Six (8.1%) patients required discontinuation of treatment, and 5 (6.8%) patients needed to change the regimen.

## Discussion

Although the benefits of early HAART onset have been described in people living with HIV in many scenarios [9–12], in the intensive care setting, such benefits are still unclear. Concerns related to absorption, drug interactions, and development of resistance, together with the absence of clinical trials, hinder decision-making. In our study, we found no benefit from the early HAART strategy for HIV-infected patients admitted to the ICU. This is the only randomized clinical trial ever to test this hypothesis. Unfortunately, the early termination of the trial may have compromised the results.

In the general population of HIV-infected patients with opportunistic infections, early onset of HAART results in slower disease progression and lower mortality, probably from earlier immune reconstitution and viral suppression. Differently from the study period, it is recommended currently to prescribe HAART for all HIV patients, regardless of CD4 count [17]. In critical patients, the decision is more difficult due to the challenges related to drug interactions, dose, absorption and adverse effects [27–29]. There are no pharmacological or pharmacokinetic data for antiretrovirals in hemodynamically unstable patients. Drug interactions are expected with antiarrhythmics, anticonvulsants, antifungals, antibiotics or sedatives [30]. In addition, most drugs are only available orally, with unpredictable absorption in critically ill patients. Subtherapeutic concentrations, even for a few days due to poor absorption or pharmacokinetic interactions, may lead to the selection of resistant virus [31]. Finally, the toxicity of antiretrovirals should be remembered. In our study, 9.6% of the patients had an adverse event, all of which necessitated a change of regimen or suspension of treatment. In another study conducted in the ICU, the incidence of an adverse event requiring a change or discontinuation of antiretrovirals was 18% [20]. The decision to initiate HAART early in this setting

should consider the risk of disease progression and these potential adverse events. Currently, this decision is made based on expert opinion.

Several retrospective studies have reported an increase in the survival of HIV-infected patients admitted to the ICU in recent decades since the advent of HAART [4, 22, 32]. The decrease in mortality may be the result of the change in the cause spectrum of admission of these patients, with non-AIDS-related diseases becoming more frequent. In addition, advances in intensive care probably contributed to improving the outcome of these patients. However, the benefit associated with the administration of HAART in the ICU is still poorly studied. A recent meta-analysis, which included 12 retrospective studies, showed a reduction in short-term mortality with the use of HAART during ICU admission [23]. This benefit persisted even after a sensitivity analysis regarding sample size, study origin and year of publication. However, no clinical trial was found in the systematic review. The conclusion of the study was restricted to the limitations of retrospective studies. It is possible that patients with the greatest potential benefit from the early use of HAART in the ICU would be patients admitted for AIDS-defining illnesses and with low CD4+ cell count, common characteristics in studies that demonstrate survival benefit [7, 33]. In our study, most patients had AIDS-defining disease as the cause of admission and CD4+ cell count < 50 cells/ml. Regardless, there was no reduction in mortality with the early onset of HAART.

Acute respiratory failure remains the main reason for ICU admission in HIV-infected patients [23, 34], although the number of patients with AIDS-defining disease and opportunistic infections admitted to the ICU has decreased considerably in the last decades in regions with access to HAART [34]. Our population presented with a high prevalence of pneumocystosis and tuberculosis as causes of respiratory insufficiency, possibly explained by the poor immunological status of patients and epidemiological factors, since these are the most reportable AIDS-defining illness in Brazil.

The risk factors for hospital mortality identified in our study were consistent with factors previously identified in the literature [31, 35]. These are factors related to organic dysfunction and not to HIV-related characteristics. Similarly, Japiassu et al. verified that sepsis was the main factor associated with hospital mortality in a prospective cohort [8]. We have identified only the use of vasopressors and the need for renal replacement therapy as factors independently associated with hospital mortality. For mortality at 6 months, in addition to these two variables, the presence of tuberculosis was an independent risk factor. Several authors have reported that HIV-related variables have had little or no impact on mortality in patients admitted to the ICU, although they are associated with long-term outcome [22, 36, 37].

The high hospital and 6-month mortality observed in our study is similar to the mortality reported in other Brazilian studies [20, 38]. These results can be explained by the disease severity of our population, both in relation to poor immune status and critical illness (elevated SAPS III), and the difficulty in accessing the public health system. Regarding immunological status, 88% of the patients presented with CD4+ < 200 cells/ml, and 71% had CD4+ <100 cells/ml. This suggests that most patients had advanced AIDS, probably contributing to the poor outcome.

Our study has some limitations. First, we did not enroll the sample size calculated, which is concerning in negative studies. However, the conditional power considering the observed effect was only 3.2%, suggesting futility as there is little evidence of a beneficial effect. Second, the study was not blinded. Although this is a limitation, we believe its impact is lessened due to the nature of the primary outcome. Finally, the study was developed in a single center, which limits the generalization of the results.

## Conclusions

Although the early termination of the study precludes definitive conclusions being made, early HAART administration for HIV-infected patients admitted to the ICU compared to late administration did not show benefit in hospital mortality or 6-month mortality. Further randomized, multicenter and larger sample size studies should be performed to better assess this hypothesis.

## Supporting information

**S1 Checklist. CONSORT 2010 checklist of information to include when reporting a randomised trial**[*].
(DOC)

**S1 Fig.**
(PNG)

**S2 Fig.**
(PNG)

**S3 Fig.**
(PNG)

**S1 File.**
(DOCX)

**S2 File.**
(DOCX)

**S1 Data.**
(CSV)

## Author Contributions

**Conceptualization:** Márcio M. Boniatti, José Augusto S. Pellegrini, Leonardo S. Marques, Josiane F. John, Luiz G. Marin, Diego R. Falci.

**Data curation:** Márcio M. Boniatti, Josiane F. John, Luiz G. Marin, Diego R. Falci.

**Formal analysis:** Márcio M. Boniatti, José Augusto S. Pellegrini, Leonardo S. Marques, Josiane F. John, Luiz G. Marin, Thiago C. Lisboa, Lucas P. Damiani, Diego R. Falci.

**Investigation:** Márcio M. Boniatti, José Augusto S. Pellegrini, Leonardo S. Marques, Josiane F. John, Luiz G. Marin, Lina R. D. M. Maito, Diego R. Falci.

**Methodology:** Márcio M. Boniatti, José Augusto S. Pellegrini, Leonardo S. Marques, Josiane F. John, Luiz G. Marin, Lina R. D. M. Maito, Thiago C. Lisboa, Lucas P. Damiani, Diego R. Falci.

**Project administration:** Márcio M. Boniatti, José Augusto S. Pellegrini, Josiane F. John, Luiz G. Marin.

**Supervision:** Márcio M. Boniatti.

**Writing – original draft:** Márcio M. Boniatti, José Augusto S. Pellegrini, Leonardo S. Marques, Josiane F. John, Luiz G. Marin, Lina R. D. M. Maito, Thiago C. Lisboa, Lucas P. Damiani, Diego R. Falci.

**Writing – review & editing:** Márcio M. Boniatti, José Augusto S. Pellegrini, Leonardo S. Marques, Josiane F. John, Luiz G. Marin, Lina R. D. M. Maito, Thiago C. Lisboa, Lucas P. Damiani, Diego R. Falci.

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
