## [Decision Letter · Decision Letter 0]

29 May 2020

PONE-D-20-06158

Early antiretroviral therapy for HIV-infected patients admitted to an intensive care unit (EARTH-ICU): a randomized clinical trial

PLOS ONE

Dear Dr. Boniatti,

Thank you for submitting your manuscript to PLOS ONE. After careful consideration, we feel that it has merit but does not fully meet PLOS ONE’s publication criteria as it currently stands. Therefore, we invite you to submit a revised version of the manuscript that addresses the points raised during the review process.

The reasons why of early termination should be adequately explained and discussed

We look forward to receiving your revised manuscript.

Kind regards,

Felipe Dal Pizzol

Academic Editor

PLOS ONE

2. Please provide additional details regarding participant consent. In the ethics statement in the Methods and online submission information, please ensure that you have specified whether consent was written or verbal/oral. If consent was verbal/oral, please specify: 1) whether the ethics committee approved the verbal/oral consent procedure, 2) why written consent could not be obtained, and 3) how verbal/oral consent was recorded. If your study included minors, please state whether you obtained consent from parents or guardians in these cases.

Reviewers' comments:

Reviewer's Responses to Questions

**Comments to the Author**

1. Is the manuscript technically sound, and do the data support the conclusions?

Reviewer #1: Yes

Reviewer #2: Yes

2. Has the statistical analysis been performed appropriately and rigorously? 

Reviewer #1: Yes

Reviewer #2: No

3. Have the authors made all data underlying the findings in their manuscript fully available?

Reviewer #1: Yes

Reviewer #2: Yes

4. Is the manuscript presented in an intelligible fashion and written in standard English?

Reviewer #1: Yes

Reviewer #2: Yes

5. Review Comments to the Author

Reviewer #1: This is a well-written study describing the results of a clinical trial to compare outcomes in ICU patients with differing onsets of ART initiation. The study addresses the data from a meta-analysis (Andrade, 2017) on the topic that indicated benefit for initiation of ART in the ICU. The study protocol is well described but recruitment to reach the calculated sample size of 172 patients per group was prematurely ended due to low enrollment with only 115 patients randomized. The failure to enroll the requisite number of patients compromises the trial design and this should be clearly stated in the abstract and throughout the manuscript. The authors need to further explain their rationale for early termination of the trial.

Minor comments:

Page 4, line 4 – the meaning of “inability to use the gastrointestinal tract” should be clarified.

Page 8 - the meaning of “…patients admitted for AIDS-defining illnesses and with poor immunoblot status…”

Reviewer #2: Overall, the methods and analyses seem to be performed well. I had a few comments with the aim of improving the description or methods.

1. (p.5) The language here suggests the trial was stopped early because of futility, but on p.6 the says the trip was stopped early due to slow recruitment. I suggest clearing this up and making sure these statements correspond with each other. Also, if there was some sort of planned interim analysis, please describe the criteria that were a priori chosen to stop for futility.

2. (p.6) Your variable selection appears to be a combination of bivariate screening and stepwise variable selection. With the sample sizes you have, stepwise variable selection procedures usually do not do a good job of finding the most appropriate model (e.g., https://doi.org/10.1002/sim.3943). Stepwise procedures and any p-value based selection have quite a bit of evidence suggesting that they are poor at selecting the appropriate variables. For a decent summary, see the link above. Bivariate screening is really just a form of stepwise selection. Sun et al. (http://dx.doi.org/10.1016/0895-4356(96)00025-X) found that bivariate screening can miss a variable that may be a confounder even when a p-value is as high as what you have. Generally, it's better to select based on more robust criteria, especially measures which assess the fit of the model or, better yet, a shrinkage-based estimator such as lasso or lars. I don't know where in SPSS this can be fit, but in R the glmnet package can do it.

3. (p.6) Although I know this is used quite a bit, I don't understand the term "independent predictors" in a multivariable model since the effect is dependent on the other predictors in the model. I recommend something along the lines of "after controlling for other factors". Also, I don't think it's "after multivariate analysis" makes sense since you are pulling the information from the model. Maybe "in the multivariate analysis".

4. It felt like survival analysis might have been a useful analysis in this trial. I'm curious if this was considered. I'm not sure the results would change, but it would seem the timing of the outcomes could have an impact.

5. Although the tables are pretty standard, I feel as though they need more descriptive titles. For instance, table 2 should say what the numbers are and what tests are used for the p-values. Probably some info on the sample as well. Table 3 should mention the statistical analyses used.

6. PLOS authors have the option to publish the peer review history of their article (what does this mean?). If published, this will include your full peer review and any attached files.

Reviewer #1: No

Reviewer #2: No

---

## [Author Response · Author response to Decision Letter 0]

19 Aug 2020

Please provide additional details regarding participant consent. In the ethics statement in the Methods and online submission information, please ensure that you have specified whether consent was written or verbal/oral. If consent was verbal/oral, please specify: 1) whether the ethics committee approved the verbal/oral consent procedure, 2) why written consent could not be obtained, and 3) how verbal/oral consent was recorded. If your study included minors, please state whether you obtained consent from parents or guardians in these cases.

The informed consent was written. We made this clearer in the manuscript. 

We note that you have indicated that data from this study are available upon request. PLOS only allows data to be available upon request if there are legal or ethical restrictions on sharing data publicly. For information on unacceptable data access restrictions, please see http://journals.plos.org/plosone/s/data-availability#loc-unacceptable-data-access-restrictions.

We made the manuscript data available through file upload (as Supporting Information file). 

Reviewer #1: This is a well-written study describing the results of a clinical trial to compare outcomes in ICU patients with differing onsets of ART initiation. The study addresses the data from a meta-analysis (Andrade, 2017) on the topic that indicated benefit for initiation of ART in the ICU. The study protocol is well described but recruitment to reach the calculated sample size of 172 patients per group was prematurely ended due to low enrollment with only 115 patients randomized. The failure to enroll the requisite number of patients compromises the trial design and this should be clearly stated in the abstract and throughout the manuscript. The authors need to further explain their rationale for early termination of the trial.

We appreciate the reviewer's comments. We made it clearer in the abstract and in the manuscript that the premature interruption of the trial may have compromised the results.

Minor comments:

Page 4, line 4 – the meaning of “inability to use the gastrointestinal tract” should be clarified.

We replaced "inability to use the gastrointestinal tract" by impossibility to use the enteral route 

Page 8 - the meaning of “…patients admitted for AIDS-defining illnesses and with poor immunoblot status…”

We replaced "…patients admitted for AIDS-defining illnesses and with poor immunoblot status…" by "patients admitted for AIDS-defining illnesses and with low CD4+ cell count"

Reviewer #2: Overall, the methods and analyses seem to be performed well. I had a few comments with the aim of improving the description or methods.

1. (p.5) The language here suggests the trial was stopped early because of futility, but on p.6 the says the trip was stopped early due to slow recruitment. I suggest clearing this up and making sure these statements correspond with each other. Also, if there was some sort of planned interim analysis, please describe the criteria that were a priori chosen to stop for futility.

We made it clearer that the trial interruption was due to a progressively slower recruitment rate. We did not perform interim analysis. 

2. (p.6) Your variable selection appears to be a combination of bivariate screening and stepwise variable selection. With the sample sizes you have, stepwise variable selection procedures usually do not do a good job of finding the most appropriate model (e.g., https://doi.org/10.1002/sim.3943). Stepwise procedures and any p-value based selection have quite a bit of evidence suggesting that they are poor at selecting the appropriate variables. For a decent summary, see the link above. Bivariate screening is really just a form of stepwise selection. Sun et al. (http://dx.doi.org/10.1016/0895-4356(96)00025-X) found that bivariate screening can miss a variable that may be a confounder even when a p-value is as high as what you have. Generally, it's better to select based on more robust criteria, especially measures which assess the fit of the model or, better yet, a shrinkage-based estimator such as lasso or lars. I don't know where in SPSS this can be fit, but in R the glmnet package can do it.

The selection of variables for the initial model was based on a combination of biological plausibility and bivariate screening. The regression model was then constructed through stepwise forward. We agree with the reviewer that stepwise forward selection may not be the most appropriate. We applied LASSO as requested by the reviewer. The R script is below:

library(glmnet)

library(ggplot2)

# Modelo LASSO Hospital Mortality

data_EARTHICU <- read.csv("~/Downloads/data_EARTHICU.csv", sep=";")

data_EARTHICU

xfactors <- model.matrix(data_EARTHICU$Hospital_mortality ~ data_EARTHICU$age + data_EARTHICU$control_treatment_group + data_EARTHICU$origin_ward + data_EARTHICU$AIDSrelated + data_EARTHICU$tuberculosis + data_EARTHICU$pneumocystosis + data_EARTHICU$cryptococosis + data_EARTHICU$saps3 + data_EARTHICU$cd4adm + data_EARTHICU$sepsis + data_EARTHICU$shock + data_EARTHICU$dialysis + data_EARTHICU$MV_adm + data_EARTHICU$ards)[, -1]

glmmod <- glmnet(xfactors, y=as.factor(data_EARTHICU$Hospital_mortality), alpha=1, family="binomial")

plot(glmmod, xvar="lambda")

coef(glmmod)[, 14]

# Modelo LASSO 6-month mortality

data_EARTHICU6 <- read.csv("~/Downloads/data_EARTHICU6.csv", sep=";")

xfactors6 <- model.matrix(data_EARTHICU6$Mortality_6_month ~ data_EARTHICU6$age + data_EARTHICU6$control_treatment_group + data_EARTHICU6$origin_ward + data_EARTHICU6$AIDSrelated + data_EARTHICU6$tuberculosis + data_EARTHICU6$pneumocystosis + data_EARTHICU6$cryptococosis + data_EARTHICU6$saps3 + data_EARTHICU6$cd4adm + data_EARTHICU6$sepsis + data_EARTHICU6$shock + data_EARTHICU6$dialysis + data_EARTHICU6$MV_adm + data_EARTHICU6$ards)[, -1]

glmmod6 <- glmnet(xfactors6, y=as.factor(data_EARTHICU6$Mortality_6_month), alpha=1, family="binomial")

plot(glmmod6, xvar="lambda")

coef(glmmod6)[, 14]

The coefficients for hospital mortality suggest the original model presented is OK, with only dyalisis and shock. LASSO regression coefficients are shown below:

> coef(glmmod)[, 14]

 (Intercept) data_EARTHICU$age 

 -0.475622602 0.006561752 

data_EARTHICU$control_treatment_group data_EARTHICU$origin_ward 

 0.000000000 0.000000000 

 data_EARTHICU$AIDSrelated data_EARTHICU$tuberculosis 

 0.000000000 0.081332622 

 data_EARTHICU$pneumocystosis data_EARTHICU$cryptococosis 

 0.000000000 0.241028316 

 data_EARTHICU$saps3 data_EARTHICU$cd4adm 

 0.000000000 -0.002027852 

 data_EARTHICU$sepsis data_EARTHICU$shock 

 0.190074265 0.884827694 

 data_EARTHICU$dialysis data_EARTHICU$MV_adm 

 0.664720062 0.000000000 

 data_EARTHICU$ards 

 0.000000000 

The LASSO coefficients for 6 mo mortality suggest that cryptococcosis should also remain in the final model with dyalisis, shock and tuberculosis. LASSO Coefficients are shown below: 

> coef(glmmod6)[, 14]

 (Intercept) data_EARTHICU6$age 

 -0.15380567 0.00000000 

data_EARTHICU6$control_treatment_group data_EARTHICU6$origin_ward 

 0.06275091 0.00000000 

 data_EARTHICU6$AIDSrelated data_EARTHICU6$tuberculosis 

 0.00000000 0.71715198 

 data_EARTHICU6$pneumocystosis data_EARTHICU6$cryptococosis 

 0.00000000 0.64094844 

 data_EARTHICU6$saps3 data_EARTHICU6$cd4adm 

 0.00000000 0.00000000 

 data_EARTHICU6$sepsis data_EARTHICU6$shock 

 0.00000000 0.69197132 

 data_EARTHICU6$dialysis data_EARTHICU6$MV_adm 

 0.98541680 0.00000000 

 data_EARTHICU6$ards 

 0.00000000

In the original logistic regression model presented, cryptococcosis bivariate OR was uncomputable and was removed from the final model due to numerical instability as this variable had an infinite estimated coefficient. This analysis will be available in supplementary material.

3. (p.6) Although I know this is used quite a bit, I don't understand the term "independent predictors" in a multivariable model since the effect is dependent on the other predictors in the model. I recommend something along the lines of "after controlling for other factors". Also, I don't think it's "after multivariate analysis" makes sense since you are pulling the information from the model. Maybe "in the multivariate analysis".

We agreed with the reviewer and made the changes as suggested.

4. It felt like survival analysis might have been a useful analysis in this trial. I'm curious if this was considered. I'm not sure the results would change, but it would seem the timing of the outcomes could have an impact.

We added the survival analysis as Supplemental File.

 exp(coef) exp(-coef) lower .95 upper .95

arv$control_treatment_group 0.8591 1.1641 0.5250 1.406

5. Although the tables are pretty standard, I feel as though they need more descriptive titles. For instance, table 2 should say what the numbers are and what tests are used for the p-values. Probably some info on the sample as well. Table 3 should mention the statistical analyses used.

We added the information in the tables as suggested by the reviewer.

---

## [Decision Letter · Decision Letter 1]

8 Sep 2020

Early antiretroviral therapy for HIV-infected patients admitted to an intensive care unit (EARTH-ICU): a randomized clinical trial

PONE-D-20-06158R1

Dear Dr. Boniatti,

We’re pleased to inform you that your manuscript has been judged scientifically suitable for publication and will be formally accepted for publication once it meets all outstanding technical requirements.

Kind regards,

Felipe Dal Pizzol

Academic Editor

PLOS ONE

Additional Editor Comments (optional):

Reviewers' comments:

Reviewer's Responses to Questions

**Comments to the Author**

1. If the authors have adequately addressed your comments raised in a previous round of review and you feel that this manuscript is now acceptable for publication, you may indicate that here to bypass the “Comments to the Author” section, enter your conflict of interest statement in the “Confidential to Editor” section, and submit your "Accept" recommendation.

Reviewer #1: All comments have been addressed

Reviewer #2: All comments have been addressed

2. Is the manuscript technically sound, and do the data support the conclusions?

Reviewer #1: (No Response)

Reviewer #2: (No Response)

3. Has the statistical analysis been performed appropriately and rigorously? 

Reviewer #1: (No Response)

Reviewer #2: (No Response)

4. Have the authors made all data underlying the findings in their manuscript fully available?

Reviewer #1: (No Response)

Reviewer #2: (No Response)

5. Is the manuscript presented in an intelligible fashion and written in standard English?

Reviewer #1: (No Response)

Reviewer #2: (No Response)

6. Review Comments to the Author

Reviewer #1: (No Response)

Reviewer #2: (No Response)

7. PLOS authors have the option to publish the peer review history of their article (what does this mean?). If published, this will include your full peer review and any attached files.

Reviewer #1: No

Reviewer #2: No

---

## [Editor Report · Acceptance letter]

10 Sep 2020

PONE-D-20-06158R1 

Early antiretroviral therapy for HIV-infected patients admitted to an intensive care unit (EARTH-ICU): a randomized clinical trial 

Dear Dr. Boniatti:

I'm pleased to inform you that your manuscript has been deemed suitable for publication in PLOS ONE. Congratulations! Your manuscript is now with our production department. 

Kind regards, 

on behalf of

Dr. Felipe Dal Pizzol 

Academic Editor

PLOS ONE